# Design and Development of Novel Composites Containing Nickel Ferrites Supported on Activated Carbon Derived from Agricultural Wastes and Its Application in Water Remediation

**DOI:** 10.3390/ma16062170

**Published:** 2023-03-08

**Authors:** Tamer S. Saleh, Ahmad K. Badawi, Reda S. Salama, Mohamed Mokhtar M. Mostafa

**Affiliations:** 1Department of Chemistry, College of Science, University of Jeddah, Jeddah 21589, Saudi Arabia; 2Civil Engineering Department, El-Madina Higher Institute for Engineering and Technology, Giza 12588, Egypt; 3Basic Science Department, Faculty of Engineering, Delta University for Science and Technology, Gamasa 11152, Egypt; 4Chemistry Department, Faculty of Science, King Abdulaziz University, Jeddah 21589, Saudi Arabia

**Keywords:** water treatment, dye degradation, nickel ferrites, activated carbon, photocatalysis, willow catkins

## Abstract

Recently, efficient decontamination of water and wastewater have attracted global attention due to the deficiency in the world’s water sources. Herein, activated carbon (AC) derived from willow catkins (WCs) was successfully synthesized using chemical modification techniques and then loaded with different weight percentages of nickel ferrite nanocomposites (10, 25, 45, and 65 wt.%) via a one-step hydrothermal method. The morphology, chemical structure, and surface composition of the nickel ferrite supported on AC (NFAC) were analyzed by XRD, TEM, SEM, EDX, and FTIR spectroscopy. Textural properties (surface area) of the nanocomposites (NC) were investigated by using Brunauer–Emmett–Teller (BET) analysis. The prepared nanocomposites were tested on different dyes to form a system for water remediation and make this photocatalyst convenient to recycle. The photodegradation of rhodamine B dye was investigated by adjusting a variety of factors such as the amount of nickel in nanocomposites, the weight of photocatalyst, reaction time, and photocatalyst reusability. The 45NFAC photocatalyst exhibits excellent degradation efficiency toward rhodamine B dye, reaching 99.7% in 90 min under a simulated source of sunlight. To summarize, NFAC nanocomposites are potential photocatalysts for water environmental remediation because they are effective, reliable, and reusable.

## 1. Introduction

The great utilization of dyes as synthetic organic compounds in various sectors became the primary source of contamination of the world’s water supplies [1]. The colors of these contaminants have been identified as one of the most significant factors influencing water quality. The existence of these colors even at a low concentration in water is extremely harmful, and the necessity for their elimination has been tested [2]. Numerous industries are commonly used dyes such as paper printing, textile, plastics, leather, etc., and nearly 95% of these dyes discharged to wastewater are absolutely considered as one of the most serious pollutants to the environment and ecosystem [3,4]. Among the most typical forms of these aromatic dyes is rhodamine B. It is a reddish violet powder that is extremely soluble in water and is widely used as a colorant in food and textiles. It has a harmful effect on both animals and humans if it is swallowed, and it could cause eye and skin irritation as well as respiratory tract irritation [5]. Additionally, it has a carcinogenic effect towards animals and human beings [6]. Given the dangers and consequences of these dyes, it was necessary to make efforts to discard these pollutants from the water using effective methods [7,8].

Numerous chemical and physical methods are commonly used to degrade/remove dyes from water, such as chemical oxidation [9,10], adsorption [11], photocatalytic degradation [12,13], and coagulation techniques [14]. Among them, photocatalytic degradation is considered as one of the most significant approaches that can aid in degrading these dangerous pollutants from wastewater. In the photocatalysis approach, the organic pollutants were totally decomposed into harmless end products, and also, the secondary hazardous by-products were not generated in this approach [15,16,17]. Photocatalysis techniques mainly depend on the highly reactive and oxidizing radicals that produced. So, it is a low-cost advanced oxidation technique with high efficiency [18]. The photodegradation efficacy of each material mainly depends on its ability to absorb light. So, using materials which have the ability to absorb light, such as semiconducting systems, is favorable and can provide better photocatalytic activity.

Semiconductor photocatalysts such as zinc oxides, nickel oxides, titania, cadmium oxides, and ferric oxides have been used in many photodegradation systems for dye degradation [19,20]. However, these materials have different drawbacks, for example, they are only effective in UV compared with the visible spectrum due to the broad band gap energies and their photo-generated electrons and holes quickly recombining before the complete photocatalytic process [21,22,23]. So, to avoid this limitation, some modifications should be introduced by adding another doping agent or by mixing different semiconductor materials to form a nanocomposite [24]. In recent times, great attention has been paid to magnetic spinel ferrites-based photo catalysts, due to their having a wide range of absorption and having a good band gap ranging from 2.0 to 3.0 eV [25]. Ferrites are a kind of spinel with the chemical formula of MFe_2_O_4_ (M = Co, Zn, Cu, Mn, Fe, Ni, etc.), based on their crystal structures and magnetic characteristics [26]. Different spinel ferrite nanocomposites such as MnFe_2_O_4_, NiFe_2_O_4_, CuFe_2_O_4_, CoFe_2_O_4_, and ZnFe_2_O_4_ have a potential photocatalytic efficiency. One of the greatest disadvantages of these spinel ferrite nanocomposites is their ability to undergo accumulation because of the aid produced by Vander Waals forces and magnetic interactions. Additionally, the metal ion dissolution in aqueous solutions caused by these nanoparticles results in additional environmental risks, limiting their relevance in photocatalytic applications [27]. Therefore, these shortages can be overcome through loading these metal ferrites on different supports such as carbon-based materials [28], mesoporous silica [29], metal organic frameworks [30], polymers [31], etc.

Among these supports, activated carbon (AC) is amorphous and has high porosity owing to the preparation method and treatment techniques. Most of the activated carbon synthesized via chemical methods is produced from fossil fuels shown to be non-eco-friendly and to require expensive techniques [32]. Thus, there has been increasing interest in the preparation of activated carbon from biomass for sustainable development. The synthesis of activated carbons (AC) from agricultural waste materials is a novel approach that has previously been reported in several studies, including works on sugar cane bagasse [33], discarded coffee beans [34], corn grains [35], banana fibers [36], and willow catkins [37,38]. Synthesized activated carbon not only focuses on environmental issues but is also a rich source of carbon precursors that have several functional groups. The presence of these functional groups having heteroatoms (as nitrogen, oxygen, sulfur, and others) in the carbon matrix enhances its activity by introducing basic and acidic characters. In addition, preparation methods are simple and environmentally friendly [39].

In recent work, WCs have been utilized as a precursor to synthesize unique activated carbons (AC) using a chemical activation method. Activated carbon was modified with different weight percentages of nickel ferrite (NiFe_2_O_4_) nanocomposites (10, 25, 45, and 65 wt.%) through a one-step hydrothermal method. The prepared nanocomposites with various content of nickel ferrites were characterized by numerous methods such as XRD, TEM, SEM, EDX, FTIR, and nitrogen sorption isotherms. The photocatalytic degradation of rhodamine B over NFAC photocatalysts was investigated in an aqueous solution. Additionally, the reusability of the catalysts was investigated.

## 2. Materials and Methods

### 2.1. Materials

To prepare nickel ferries nanoparticles, nickel nitrate (Ni(NO_3_)_2_·6H_2_O, 98.0%, Sigma-Aldrich, St. Louis, MO, USA), iron nitrate (Fe(NO_3_)_3_.9H_2_O, 99.9%, Sigma-Aldrich), and Ammonium hydroxide (NH_4_OH, 33%) were used as received as raw materials. Additionally, willow catkins (WC) were collected in April from the Nile River in the Damietta Governorate, Egypt. Hydrochloric acid (HCl, 30–36%) and potassium hydroxide (KOH) were purchased from Alfa Aesar.

### 2.2. Synthesis of AC from WCs

AC derived from WCs was synthesized according to a previous study by Wang et al. [37]. Initially, a certain quantity of willow catkins were repeatedly washed with distilled water to eliminate any soil particles and contaminants on their surfaces. After that, they were dried in an electrical oven at 120 °C overnight. Then, 100 mL of WC and KOH were combined in a 1:2 mass ratio of deionized water for 1 h at room temperature, then water evaporated at 120 °C in electrical oven overnight. The dried sample was calcined at 600 °C for 3 h under N_2_ flow of 30 mL·min^−1^. Finally, the produced powders were then rinsed with HCl (1.0 M) and washed with deionized water until the pH of filtrate reached 7, and then the precipitate was dried at 120 °C overnight to form the activated carbon (AC).

### 2.3. Preparation of Different Weight Contents of NiFe_2_O_4_/AC

Briefly, desired amounts of iron nitrate and nickel nitrate with molar ratio of 2:1 were each dissolved in 20 mL distilled water. After mixing, 2.0 g of AC were added to the solution and sonicated for 1 h at 37 kHz using Elmasonic equipment to produce the optimum dispersion of ferrites nanoparticles on the surface of AC. After that, 10 mL of ammonia solution (33%) was slowly added to the prepared suspension until the pH of the suspension reached to 10. After that, the as-prepared suspension was moved to a Teflon-lined hydrothermal autoclave and heated in an electrical oven for 6 h at 180 °C. The produced precipitate was filtered and washed with deionized water and then dried at 120 °C overnight. Finally, the resultant powders were calcined at 500 °C to produce nickel ferrites supported on activated carbon (NFAC). The produced NCs were labelled as xNFAC, where x is the loaded amount of nickel ferrites nanoparticles that was calculated to be 10, 25, 45, and 65 wt.%.

### 2.4. Characterization

X-ray diffraction patterns of the prepared catalysts were carried out through XRD-7000 Shimadzu-Japan using a Cu K α radiation X-ray source (λ = 1.5406 °A) which operated at 30 mA and 40 kV. The scanning was completed at a 2θ angle from 5 °A to 80 °A, with a step size of 0.02 and a step time of 2 s, while the functional groups and the chemical bonds were carried out using a FTIR spectrophotometer (PerkinElmer-Spectrum 2B, USA). The FTIR spectrum is recorded between 400 cm^−1^ and 4000 cm^−1^ after mixing the prepared nanocomposites with 0.1 g KBr in 5 mm diameter self-supporting discs. Additionally, the surface morphology and chemical composition of x NFAC were investigated by scanning electron microscopy (SEM) (JEOL, JSM-IT200) analysis and its associated unit energy dispersive X-ray spectroscopy (EDX). On the other hand, transmission electron microscopy (TEM) was used to display the morphology and the crystallite size of the prepared nanoparticles using TEM-JEOL, JEM-2100. Nitrogen sorption measurement was applied to determine the pore diameter and the surface area of the as-synthesized nanocomposites using BELSORP-mini II instrument.

### 2.5. Photocatalytic Degradation Measurement

The photocatalytic degradation of rhodamine B was performed using different wight contents the as-synthesized photocatalysts in an aqueous solution. The photocatalytic reaction was carried out using xenon lamp (Xe with 150 W) as a simulated sunlight irradiation without using radiation filter, and the reactor was enclosed by a water-cooling system. All tests were achieved at room temperature. In a typical measurement, 50 mg of NFAC photocatalyst was transferred into 50 mL of aqueous solution with 10 ppm of rhodamine B dye, and then the systems were transferred to a photoreactor. In this study, the reactor was stirred in the dark for 30 min until adsorption–desorption equilibrium for rhodamine B attained. After that, the lamp turned on, and the mixture was stirred and irradiated; then, 2 mL from the resultant mixture was centrifuged at different time intervals before absorbance measurements using Shimadzu, MPC-2200 UV-Vis spectrophotometer. The photodegradation percentages or removal rate (D%) of rhodamine B were calculated from Equation (1) [40]:(1)$$D\%\;=\left(\frac{\mathrm{Co}-\mathrm{Ct}}{\mathrm{Co}}\right)\times 100\%$$
where *C_o_* and *C_t_* are the concentration of initial dye and after irradiation time (*t*), respectively.

## 3. Results and Discussion

### 3.1. X-ray Diffraction Analysis

XRD diffractions of pure activated carbon (AC) derived from WCs and different weight contents of NiFe_2_O_4_ nanocomposites are given in Figure 1. Figure 1a shows that AC has two broad peaks at 2θ with corresponding diffraction planes equal to 26.1 (002) and 42.9° (100), respectively. These results reveal that AC is amorphous in nature, which is in alignment with the PDF number of standard card of 41–1487 [41,42], while Figure 1b–e display the XRD of different NiFe_2_O_4_ supported on AC, which shows that there are new sharp peaks observed when the weight content of nickel ferrites increased from 10.0 wt.% to 65.0 wt.% and the intensity of these peaks was increased as the content of nanoparticles increased. These peaks were observed in all the modified samples, which were observed at diffraction angles (2θ) with corresponding planes at 30.4° (220), 35.9° (311), 37.9° (222), 43.4° (400), 54.5° (422), 57.4° (511), and 63.5° (440), respectively [43,44]. These results are well fitted with the PDF number of standard card of 10-0325 [45]. It clearly shows that the NiFe_2_O_4_ nanoparticles are successfully formed and supported on activated carbon. The average crystallite size of nickel ferrites nanoparticles was calculated with the help of Scherrer’s equation, and the results indicate that the crystallite size of NiFe_2_O_4_ nanoparticles was increased with the increase in the content of nickel ferrites from 9.43 nm at a low concentration of nickel ferrites (10NFAC) to 15.79 nm at a higher concentration (65NFAC), as displayed in Table 1. These increases in the crystallite size may be related to the agglomeration of nickel ferrites on the surface of the activated carbon, which could increase its size.

### 3.2. FT-IR Spectroscopy

Figure 2 displays the FTIR spectra of the prepared NCs. For each spectrum, a large band at 3421 cm^−1^ was seen, which corresponds to the O–H stretching vibration mode of carboxyl, hydroxyl, or surface-adsorbed water, suggesting the presence of free hydroxyl on the carbon surface [46]. The large peak seen in the activated carbon at roughly 2000–2300 cm^−1^ might be attributed to an asymmetrical stretch of O=C=O trapped inside the pores of the activated carbon during the calcination process [47]. The peak observed at 1634 cm^−1^ could be related to asymmetric stretching vibration of C=O of the carboxylic group [48] or H-O-H vibrational bending of water molecules [49]. The broad peak observed at wavenumber of 1009 cm^−1^ could be connected to the activated carbon’s skeleton C-O stretching vibrations. [50]. On the other hand, the spectra of 10NFAC, 25NFAC, 45NFAC, and 65NFAC composites reserved all of the characteristic peaks existing in pure AC, revealing that the types of functional groups thereon were unchanged. Figure 2b–e show that two distinctive peaks were seen in nickel ferrite structures that were derived from oxygen bonding. The first of these peaks, which was associated with the stretching vibration of the tetrahedral structure of metal–oxygen interaction, was often seen in the 550–650 cm^−1^ range. The second, relating to the metal–oxygen bond in the octahedral structure mentioned in the preceding literature, was seen at a range of 400 to 450 cm^−1^ [51,52,53].

### 3.3. Textural Properties Measurements

Nitrogen sorption isotherms of AC, 10NFAC, 25NFAC, 45NFAC, and 65NFAC composites were measured at liquid nitrogen temperature (−196 °C) and are illustrated in Figure 3. All the prepared composites show a high N_2_ uptake at low relative pressure, which reveals that all the samples displayed type I isotherms, which is attributed to microporous materials with a pore size less than 2 nm, according to the IUPAC classification. As displayed in Figure 3, the sharp nitrogen gas uptake in all the as-synthesized photocatalysts at a low relative pressure reveals that most of the exposed surface resides inside the micropores. These results were confirmed by the pore size distribution curves that are displayed in Figure 4. According to the pore size distribution curves, AC displayed only one peak having a dimension at 1.87 nm. On the other hand, the modified activated carbons exhibited a bimodal pore size distribution with pore diameters less than that of the pure AC. The specific surface area (SBET), pore volume, and pore diameter were measured and are displayed in Table 1. AC displayed an excellent surface area which reached 1834 m^2^.g^−1^ then decreased after modification with NiFe_2_O_4_ until reached 531 m^2^.g^−1^ at 65 NFAC, which may be due to the accumulation of nickel ferrites on the surface of AC that could block the pore or decrease its size [54].

### 3.4. Transmission Electron Microscopy (TEM) Images

The morphology and crystallite size of pure AC, 10NFAC, 25NFAC, 45NFAC, and 65NFAC composites are illustrated using TEM images and displayed in Figure 5. The TEM image of AC (Figure 5a) reveals that AC derived from WC showed amorphous nano-structures related to the graphite layers as previously reported in the literature [37]. After modification with different content of NiFe_2_O_4_, different spherical nanoparticles were detected on the surface of AC, as displayed in Figure 5b–e. According to the figures, the spherical nanoparticles count was increased with an increase in the content of nickel ferrites, and also, the agglomeration was increased with an increase in the contents. In addition, the particle size of nickel ferrites in 10NFAC, 25NFAC, 45NFAC, and 65NFAC composites was measured from the TEM instrument, and its values were 8.14, 12.17, 13.78 and 15.07 nm, respectively, which matched the data obtained from the XRD patterns. In addition, the particle size distribution histograms for as-synthesized photocatalysts were measured by using Image-J software version 1.8.0, and the results are displayed in Table 1 and Figure 6. The average crystallite sizes 10NFAC, 25NFAC, 45NFAC, and 65NFAC composites are equal to 8.14, 12.17, 13.78, and 15.07 nm, respectively, as seen in Figure 6, and these results were consistent with previously published publications revealing that the crystallite size of ferrites nanoparticles ranged from 9 to 50 nm [55,56,57]. These results are well fitted with the data obtained from XRD patterns.

### 3.5. SEM Images and EDX Analysis

The morphology and elemental composition of the as-synthesized photocatalysts were measured using scanning electron microscopy equipped with X-ray micro analysis and are displayed in Figure 7 and Figure 8, respectively. Figure 7 displays the SEM images of pure AC, 10NFAC, 25NFAC, 45NFAC, and 65NFAC composites, which reveals that the SEM images of AC has a sheet-like structure as confirmed by TEM images, while other modified samples display that some aggregates were observed on the surface of AC, which increase with an increase in the contents of NiFe_2_O_4_. These results were well fitted with the data observed from TEM and XRD patterns.

Energy-dispersive X-ray analysis (EDX) was achieved to determine the elemental composition in the as-synthesized photocatalysts and is displayed in Figure 8. Pure AC displays two sharp peaks related to (C) and oxygen (O) and some traces of sulfur (S), silicon (Si), and sodium (Na) in its structure. These traces could be related to the elements exist in willow catkins plants, which also observed in all the prepared composites [37]. After modification with NiFe_2_O_4_, there are another two peaks observed with different carbon intensities which are attributed to nickel (Ni) and iron (Fe). In addition, the peak intensity of oxygen atom was increased after increasing the content of nickel ferrites.

### 3.6. UV-Vis Diffuse Reflectance Spectroscopy

The UV-Vis absorption spectra of the produced nickel ferrites supported on AC are shown in Figure 9. It is obvious that pure AC exhibits a strong peak at a wavelength between 300 and 410 nm, which is associated with the AC. The figure also shows another distinctive peak at a wavelength between 350 and 550 nm, which was linked to the nickel ferrites surface plasmon resonance (SPR) band. Additionally, the AC sample exhibits high UV-Vis absorption below a band edge of around 530 nm, whereas the x NFAC samples show a dispersed absorption band in the visible light area. The intensity of this unique peak grew constantly as the quantity of nickel ferrites on AC increased, which may be connected to the characteristic SPR band of immobilized ferrites over AC. Table 1 shows the band gap energies of all photocatalysts computed using Tauc’s equation. The band gaps of AC, 10 NFAC, 25 NFAC, 45 NFAC, and 65 NFAC samples were 2.56, 2.01, 1.74, 1.55, and 1.63 eV, respectively, as listed in Table 1. The results showed that increasing the quantity of nickel ferrites lowered the band gap until it reached a minimum value of 45 NFAC and then grew again. This rise might be attributed to the agglomeration of nickel ferrites nanoparticles, which prevent electron excitation from the valence to conduction band.

### 3.7. Photocatalytic Activity

#### 3.7.1. Photodegradation of Rhodamine B

The photocatalytic activities of all the as-synthesized composites were measured through the photodegradation of RB under simulated sunlight irradiation in an aqueous solution and are shown in Figure 10. To reveal the photodegradation of the prepared samples, The photocatalytic activity was studied without using any photocatalyst, and the results displayed that there is no significant degradation detected in the case where there is no catalyst. Figure 10 shows that the catalytic reaction was tested in dark and light to reveal the significant role of adsorption process in the degradation of dye removal in the aqueous solution. The results displayed that activated carbon has a higher adsorption capacity compared with other photocatalysts with different weight contents of nickel ferrites, which could be related to the high surface area of the AC compared with other samples. This high adsorption capacity could affect the photodegradation activity due to the adsorbed dyes covering most of the active sites of the photocatalyst, which prevent the contact of visible light with the photocatalyst surface, so the ability of the photocatalyst to create electron–hole pairs decreases [58]. According to Figure 10, the photodegradation activity of rhodamine B gradually increased with an increase in the weight percentage of NiFe_2_O_4_ until it reached the maximum performance at 45NFAC, and then it decreased again. The increase in the photocatalytic activity of AC after modified with different weight percentages of NiFe_2_O_4_ could be attributed to the increase in oxygen vacancies, which has a significant role in reducing the recombination rate of photogenerated e^–^/h^+^ [59]. These results indicate that the amount of NiFe_2_O_4_ was the crucial factor in enhancing and reducing the photodegradation of RB.

The Langmuir–Hinshelwood first-order kinetic model was performed to recognize the degradation performance of rhodamine b over NFAC composites and examined through the following formula:(2)$$ln\;\left(\frac{C_{o}}{C_{t}}\right)=kt$$
where *t* and *k* are time and rate constants of degradation reaction. A straight line was obtained when ln (*C_o_*/*C*) was plotted versus time (*t*), as shown in Figure 11. The values of the rate constant of degradation time and its corresponding correlation coefficient for all the prepared composites were calculated from the slope of straight line and are summarized in Table 2. According to these values, the photodegradation kinetics of rhodamine B obeys the pseudo 1st order model, and the photodegradation kinetics depend only on the RB concentrations at a fixed amount of the photocatalyst. In addition, the k values were increased with an increase in the contents of NiFe_2_O_4_, and 45NFAC displayed the highest degradation rate among the tested photocatalysts.

The reaction mechanism of the photocatalysis of rhodamine B could be illustrated as the following: when the nickel ferrites loaded on the activated carbon composites are exposed to solar light, the electrons in the valence band of NiFe_2_O_4_ are excited to the conduction band. These electrons react with oxygen in the liquid forming O_2_^−•^. Additionally, the hydrogen ion formed from the dissociation of a water molecule reacts with O_2_^−^ to form ^•^OH radicals. These ^•^OH radicals have the main role for the degradation of rhodamine B dye in aqueous solution, as displayed in the Equations (3)–(7) [60,61]. By releasing more hydroxyl radicals, the photocatalysis proceeds at a faster rate [62]. In addition, the photogenerated hole transfer takes place from the valence band of NiFe_2_O_4_ to the valence band of the activated carbon. This suggests that the photogenerated electrons and holes in the heterojunction were efficiently separated [63,64,65]. Additionally, according to the previous literature, H-NMR spectra was performed on the degradation of rhodamine B over TiO_2_ [66] and WO_3_/MoCl_5_ [67], being the same as our prepared NFAC system, and the main peaks related to the aromatic hydrogen and ethyl group atoms disappeared. In addition, new peaks were observed at δ1.15–1.25 ppm and 3.0–3.1 ppm, which were attributed to the signals of –CH_3_.
(3)$${\mathrm{NiFe}}_{2}O_{4}/AC+hv\;\to \;NiFe_{2}O_{4}/AC\;\left(h^{+}+e^{-}\right)$$
(4)$$e^{-}+O_{2}\to O_{2}^{-•}\;\;$$
(5)$$e^{-}+O_{2}+2H^{+}\to \;H_{2}O_{2}\;$$
(6)$$H_{2}O_{2}+O_{2}^{-•}\to OH^{-}+OH^{•}+O_{2}$$
(7)$$OH^{•}+Rhodamine\;B\to degradation\;products$$

#### 3.7.2. Reusability of the Catalyst

The stability of the as-synthesized composites was introduced by carrying out reusability test over 45NFAC composites and is illustrated in Figure 12. The figure shows the photodegradation activity of RB over 45NFAC for five runs under the same photo-reaction conditions. In every cycle, the recycled photocatalyst was centrifuged to be separated from dye solution and then immersed for 120 min in ethanol, and then it was washed with distilled water and dried in a vacuum oven at 80 °C for 12 h. The results show that even after five cycles, the photocatalytic performance does not significantly decline, demonstrating the outstanding stability and reusability of the photocatalysts in their as-synthesized form. The stability of the 45 NFAC sample was examined by comparing the XRD characteristic peaks before and after the reusability test, as displayed in Figure 13. The figure also reveals a little variation in peak intensity, demonstrating that the 45 NFAC catalyst is quite stable even after being recycled four times. These findings confirm not only that NFAC nanocomposites can be used for sustainable photodegradation of dyes, but also that they have outstanding reusability, making them appropriate as a stable and friendly environment composite for photodegradation of organic pollutants. After a fifth cycle, NFACs, now referred to as sludge, are recovered (dye is separated from the composite) and can be used again in diverse electronic and biomedical applications including gas sensors, magnetic storage systems, microwave devices, and site-specific drug delivery [68].

## 4. Conclusions

Herein, we have established excellent photocatalytic activity of nickel ferrites nanocomposites supported on activated carbon derived from willow catkins. The prepared nanocomposites with different weight percentage of nickel ferrites were applied to photocatalytic degradation of an organic rhodamine dye pollutant in aqueous solution. The results showed that the photodegradation performance of the as-synthesized photocatalysts was enhanced after modification with NiFe_2_O_4_. The photocatalytic activity was increased with increasing ferrite contents until reaching the maximum performance at 45 NFAC, and then it decreased again, which may be due to the accumulation of NiFe_2_O_4_ on the surface of the activated carbon. Among the synthesized photocatalysts, 45 NFAC composite displayed the highest photodegradation efficiency for the degradation of rhodamine B dye due to the excellent distribution of nickel ferrites on AC and to the increase in oxygen vacancies, which has a significant role in reducing the recombination rate of photogenerated e^–^/h^+^. The results displayed that NFAC nanocomposites are efficient and stable photocatalysts which act as a promising photocatalyst for the environmental remediation of water. The real (large-scale) applications of the investigated NFAC nanocomposites for real heavily polluted wastewater, including ease of operation, cost evaluation, material separation, and reuse, are still of interest and need further research.

## Figures and Tables

**Figure 1 materials-16-02170-f001:**
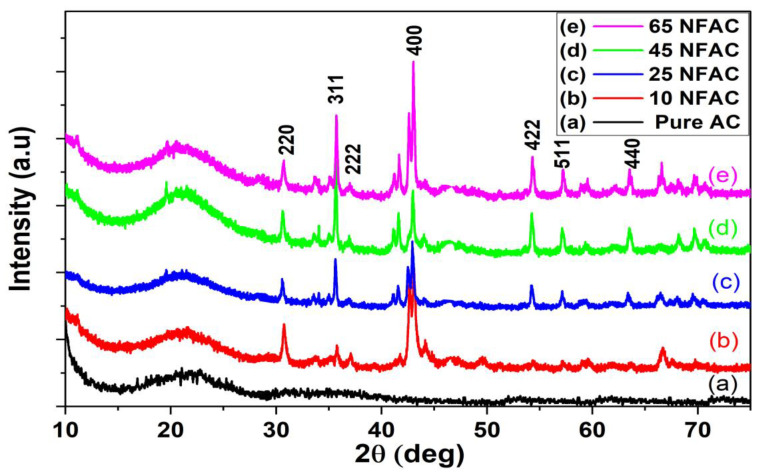
XRD patterns of (**a**) pure AC, (**b**) 10NFAC, (**c**) 25NFAC, (**d**) 45NFAC, and (**e**) 65 NFAC.

**Figure 2 materials-16-02170-f002:**
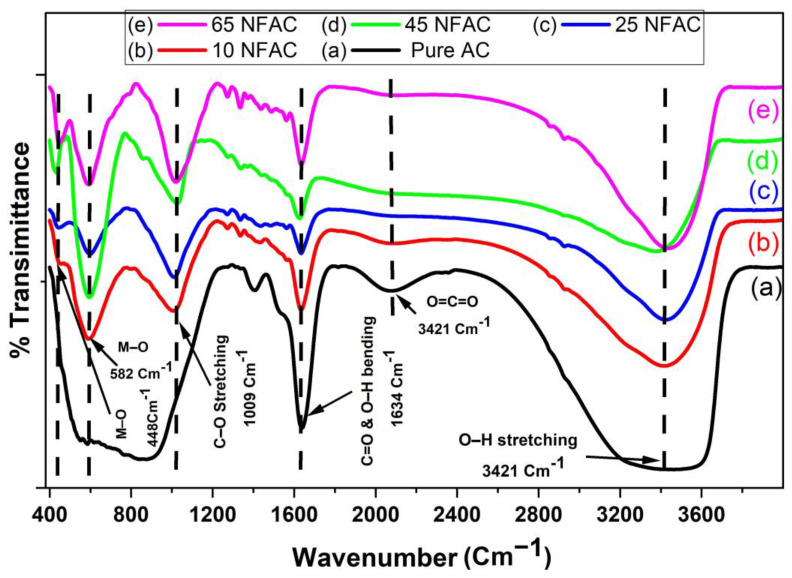
FTIR Spectra of (**a**) pure AC, (**b**) 10NFAC, (**c**) 25NFAC, (**d**) 45NFAC, and (**e**) 65 NFAC.

**Figure 3 materials-16-02170-f003:**
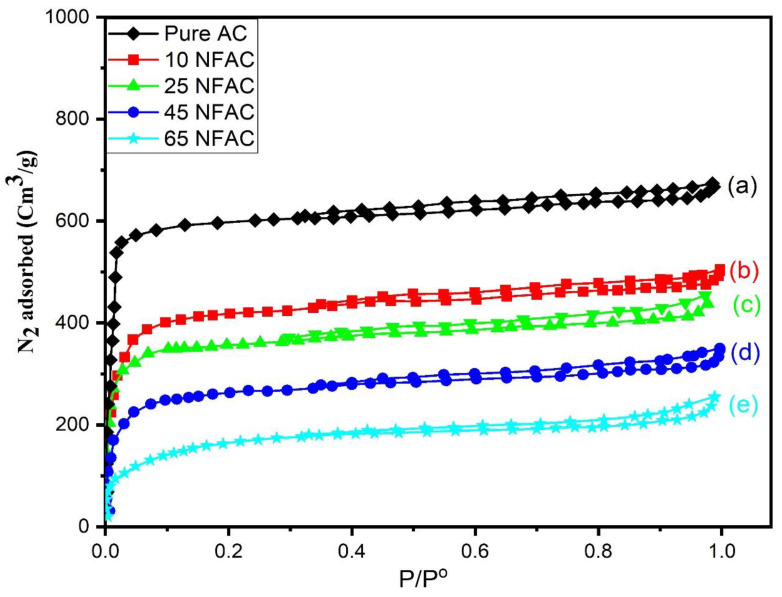
Nitrogen adsorption–desorption isotherms of (**a**) pure AC, (**b**) 10NFAC, (**c**) 25NFAC, (**d**) 45NFAC, and (**e**) 65 NFAC.

**Figure 4 materials-16-02170-f004:**
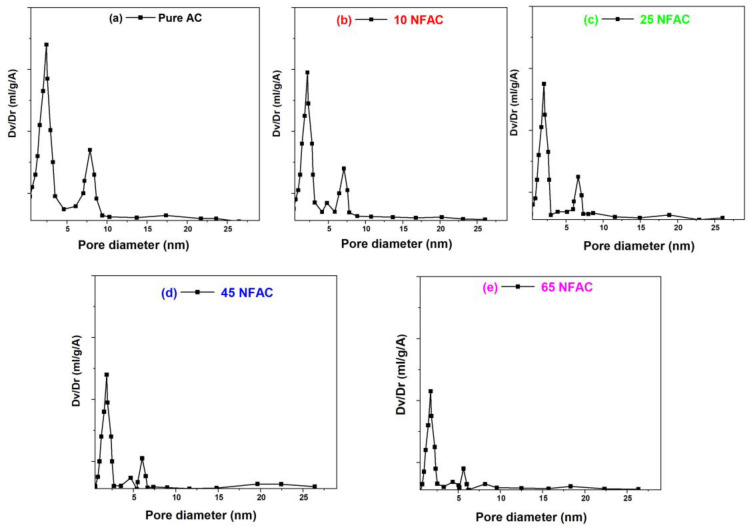
Pore size distribution curves of (**a**) pure AC, (**b**) 10NFAC, (**c**) 25NFAC, (**d**) 45NFAC, and (**e**) 65 NFAC.

**Figure 5 materials-16-02170-f005:**
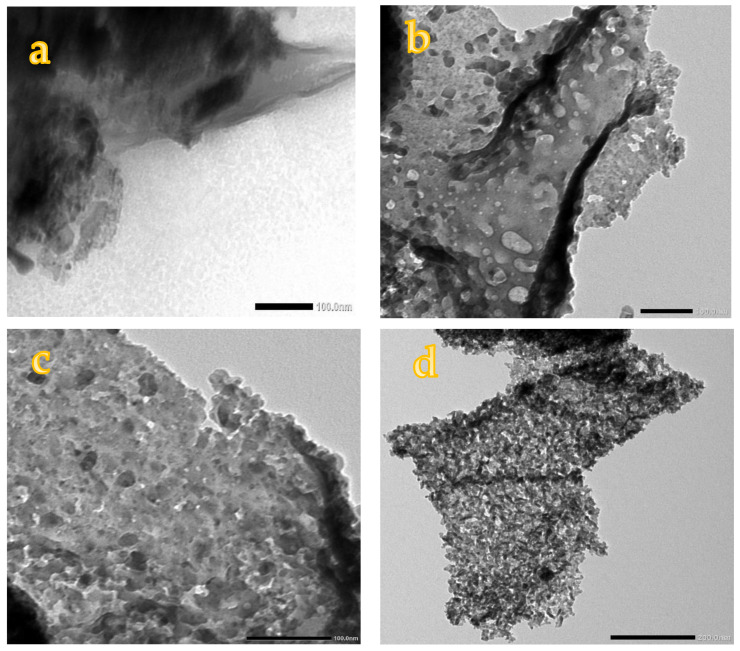
TEM images of (**a**) pure AC, (**b**) 10NFAC, (**c**) 25NFAC, (**d**) 45NFAC, and (**e**) 65 NFAC.

**Figure 6 materials-16-02170-f006:**
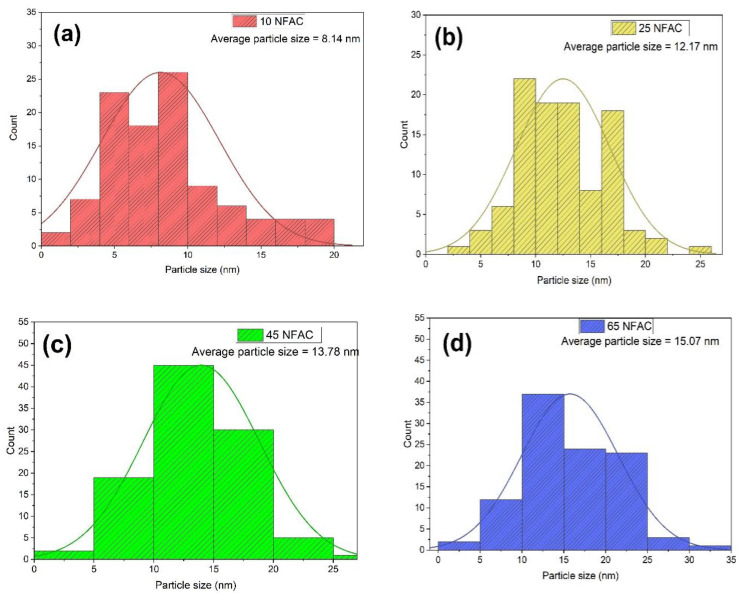
Particle size distribution histograms of (**a**) 10NFAC, (**b**) 25NFAC, (**c**) 45NFAC, and (**d**) 65 NFAC.

**Figure 7 materials-16-02170-f007:**
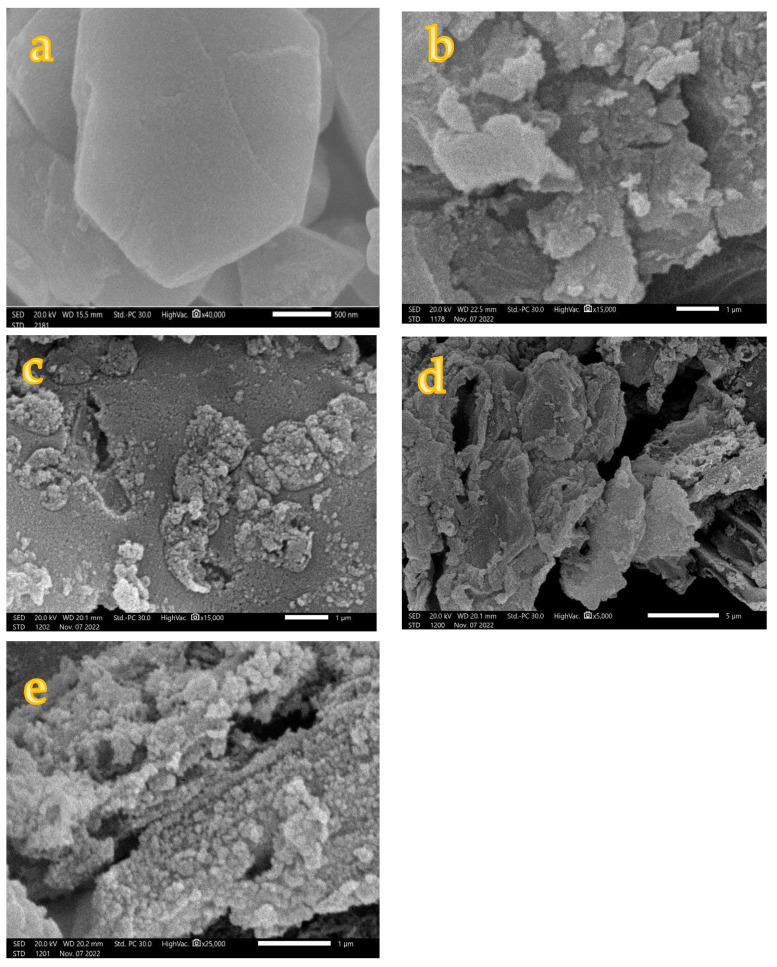
SEM images of (**a**) pure AC, (**b**) 10NFAC, (**c**) 25NFAC, (**d**) 45NFAC, and (**e**) 65 NFAC.

**Figure 8 materials-16-02170-f008:**
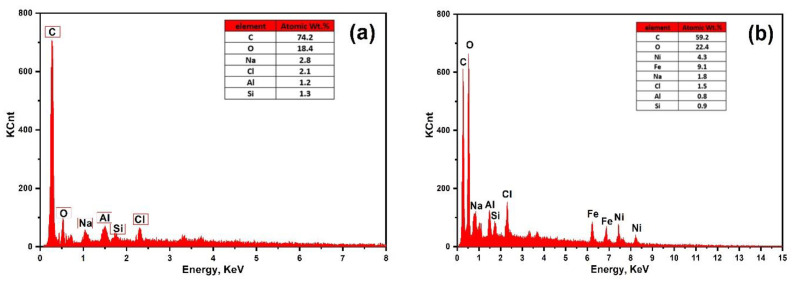
EDX analysis of (**a**) pure AC, (**b**) 10NFAC, (**c**) 25NFAC, (**d**) 45NFAC, and (**e**) 65 NFAC.

**Figure 9 materials-16-02170-f009:**
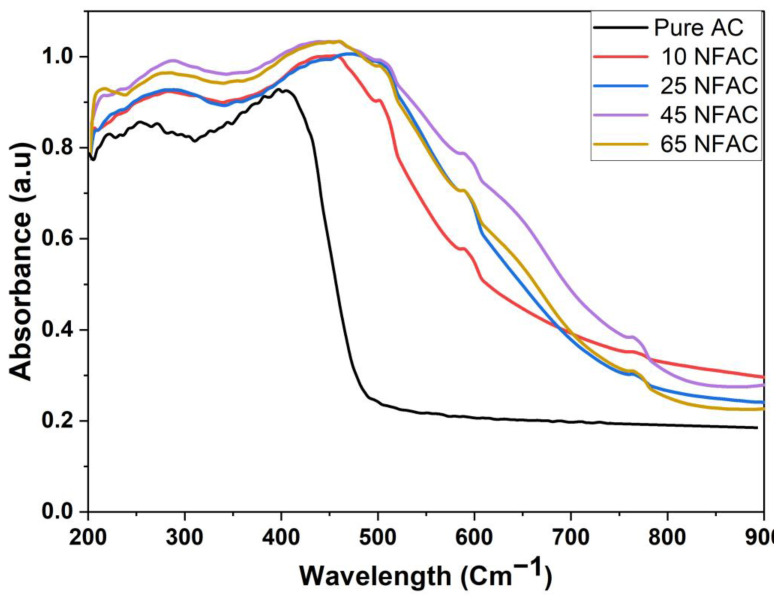
UV-Vis Diffuse Reflectance Spectroscopy of nickel ferrites with varying weight percentages in both pure and modified AC.

**Figure 10 materials-16-02170-f010:**
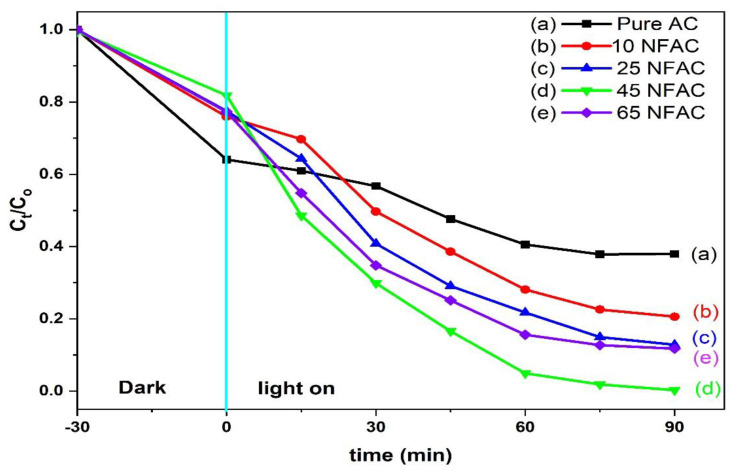
Photocatalytic degradation of rhodamine B over pure and modified activated carbon by different weight contents of NiFe_2_O_4_ nanoparticles vs. irradiation time.

**Figure 11 materials-16-02170-f011:**
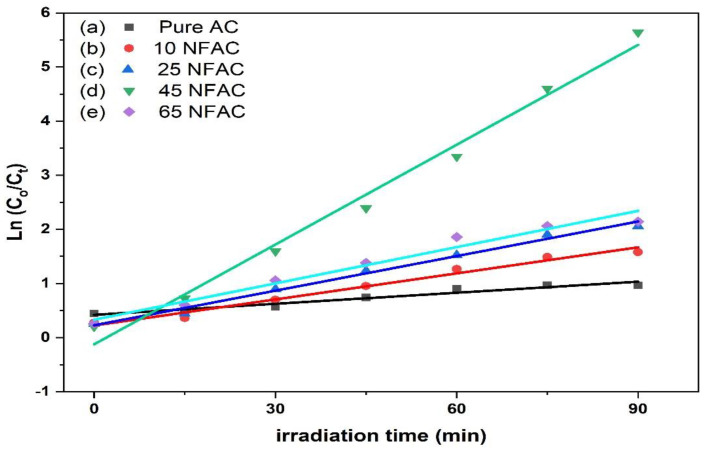
The pseudo first-order kinetic of photocatalytic degradation of rhodamine B over pure and modified activated carbon.

**Figure 12 materials-16-02170-f012:**
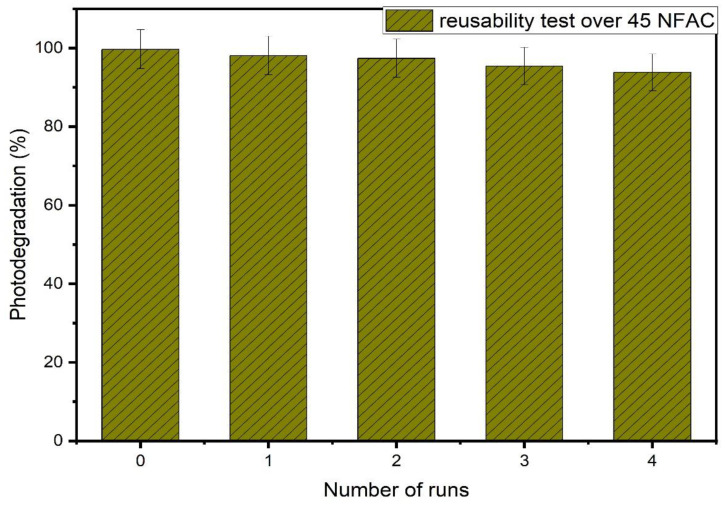
Effect of reusing times of photocatalyst on photocatalytic degradation of RB over 45NFAC nanocomposites.

**Figure 13 materials-16-02170-f013:**
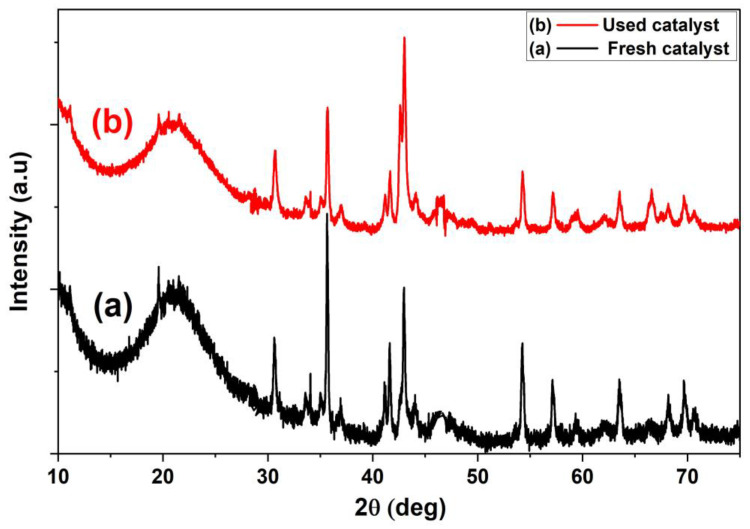
Effect of reuse on the structure of 45 NFAC catalyst using XRD pattern.

**Table 1 materials-16-02170-t001:** Textural properties and crystallite size of pure and modified AC.

Sample	S_BET_ m^2^.g^−1^	V_T_ (mL. g^−1^)	Pore Diameter (nm)	Band Gap (eV)	Crystallite Size	
					XRD (nm)	TEM (nm)
Pure AC	1834	0.89	1.87	2.56	---	---
10NFAC	1294	0.82	1.75	2.01	9.43	8.14
25NFAC	1108	0.74	1.67	1.74	13.14	12.17
45NFAC	822	0.68	1.51	1.55	14.18	13.78
65NFAC	531	0.51	1.09	1.63	15.79	15.07

**Table 2 materials-16-02170-t002:** Correlation coefficients and rate constants for rhodamine B photodegradation.

Sample Name	RB	
	K_1_	R^2^
AC	0.0068	0.93736
10NFAC	0.0161	0.96795
25NFAC	0.0213	0.98761
45NFAC	0.0615	0.98482
65NFAC	0.0223	0.96588

## Data Availability

The data presented in this study are available in the article.

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
