# Peer review of "Design and Development of Novel Composites Containing Nickel Ferrites Supported on Activated Carbon Derived from Agricultural Wastes and Its Application in Water Remediation"

_materials, 2023, doi:10.3390/ma16062170_

Round 1
Reviewer 1 Report
Manuscript Title: Design and development of novel composites containing nickel ferrites supported on Activated carbon derived from agricultural wastes and its application in water remediation
Journal Title: Materials
Authors: Tamer S. Saleh1, Ahmad K. Badawi, Reda. S. Salama, Mohamed Mokhtar M. Mostafa
Manuscript ID: materials-2217529
The article can be published in the Materials after major revision. Authors should consider the comments and respond to each of the comments presented below:
- Experimental part. Please indication initial concentration of Rhodamine B. And what is the average concentration of dyes in wastewater that should be clean up?
- Experimental part. “…then mixed together and 2.0 gram of AC were 125 added to the solution and sonicated for 1 hr.” Please specify power of sonication, which ultrasonic bath did you use?
- Experimental part. “…with 0.1g KBr in 30nm diameter self-supporting discs…” Please check and correct the diameter of the disc.
- Results and Discussion «Scherrer's equation and the results indicates that the average crystal size”. It should be “crystallite size”.
- Results and Discussion “These increase in the crystal size may be related to the agglomeration of nickel ferrites on the surface of activated carbon that could raise its size. “ It should be corrected. “crystallite size” is not the same as “particle size”. If the authors mean particle agglomeration.
- FTIR spectroscopy. Please indicate all main peaks presented in the Figure 2. It will be helpful for readers. Please indicate broad peak at around 2000-2300 cm-1 of pure AC.
- “The peak observed at 1634 cm- 1 could related to 193 asymmetric stretching vibration of C = O of the carboxylic group” it also can be attributed to the water presents. Please discuss it.
- From Table 1, it is seen that one particle of NiFe2O4 consists of one crystalline, isn’t it? Please compare obtained results with previous published.
- Figure 4. Please reconstruct the graph. It is hardly seen the changes.
- Authors did not performed experiments on water purification using real wastewater. It is recommended to performed experiments on real wastewater, since there are many components that can significantly affect both catalytic properties and contaminate the catalyst
- Authors should discuss how prepared composites can be used in real experiments. Сan the composite contaminate water, and does it make it dangerous for human consumption. How can the composite be removed from the water after purification?
Author Response
The authors are grateful to the reviewers for their corrections that have been used to improve the quality of the manuscript. The comments below have also been used to update the manuscript and we are grateful to the editor and reviewers for their meaningful contributions. The response to each comment was typed in highlighted color.

Reviewer 2 Report
Dear authors,
There are some details that I believe require correction or clarification:
-Section 2.2. How has the number of moles in WC been calculated if it is not a compound but a plant? Also, a 2:1 ratio would not be equimolecular. This word should be reserved for the 1:1 ratio.
-Paragraph 3.1. I think "Fig. 1 (a-d)" should be "Fig. 1 (b-e)".
-Section 3.2. I can't see the peaks at 554 and 673 well in the figure. But I can clearly see new peaks at 450, 600 and 900 cm-1. Can you check the wave numbers again? Or better still, point out these peaks in the figure.
Author Response

(The authors gave the same response as above.)

Reviewer 3 Report
The manuscript entitled “Design and development of novel composites containing nickel ferrites supported on Activated carbon derived from agricultural wastes and its application in water remediation” submitted by Tamer S. Saleh et al. for publication in Materials. The prepared materials were examined by XRD, electron microscopy, FTIR and BET techniques. The application part has also been covered significantly. But still there are some points to be fixed in the manuscript:
1. FTIR data should be indexed with proper functional groups (Figure 2)
2. What is the optical band gap of the prepared material? Experimental evidences are needed?
3. What are the degraded products? An appropriate reaction mechanism of the degraded products should be discussed.
4. Stability tests are very important for any material performing as a photo-catalyst. Any morphological robustness, chemical compositional or oxidation state changes occur after photo-catalysis or not? Need experimental evidences in support of stability.
Author Response

(The authors gave the same response as above.)

Round 2
Reviewer 3 Report
The main concern is still missing regarding the dye degraded products with an appropriate reaction mechanism based on the degraded products. GC-MS or NMR techniques can be used to investigate the dye degraded products. Author can follow the following work based on the degradation pathways of other dyes published by others. Only for examples;
https://www.sciencedirect.com/science/article/pii/S0926337300002769; https://www.sciencedirect.com/science/article/pii/S2214714422003099; https://pubs.rsc.org/en/content/articlelanding/2017/nj/c7nj02085f/unauth;
FTIR bands in Figure 2 are still not indexed.
Author Response
Dear Reviewer 3: We deeply appreciated your valuable comments and constructive suggestions which are much beneficial to improving the scientific quality of this manuscript. We accepted all the suggestions and comments and have performed a careful revision. We hope the revision could be satisfactory.
